

# *aaquetzalli* is required for epithelial cell polarity and neural tissue formation in *Drosophila*

Miguel A. Mendoza-Ortíz, Juan M. Murillo-Maldonado and Juan R. Riesgo-Escovar

Developmental Neurobiology and Neurophysiology, Instituto de Neurobiología, Universidad Nacional Autónoma de México, Querétaro, México

## ABSTRACT

Morphogenetic movements during embryogenesis require dynamic changes in epithelial cell polarity and cytoskeletal reorganization. Such changes involve, among others, rearrangements of cell-cell contacts and protein traffic. In *Drosophila melanogaster*, neuroblast delamination during early neurogenesis is a well-characterized process requiring a polarized neuroepithelium, regulated by the Notch signaling pathway. Maintenance of epithelial cell polarity ensues proper Notch pathway activation during neurogenesis. We characterize here *aaquetzalli* (*aqz*), a gene whose mutations affect cell polarity and nervous system specification. The *aqz* locus encodes a protein that harbors a domain with significant homology to a proline-rich conserved domain of nuclear receptor co-activators. *aqz* expression occurs at all stages of the fly life cycle, and is dynamic. *aqz* mutants are lethal, showing a disruption of cell polarity during embryonic ventral neuroepithelium differentiation resulting in loss of epithelial integrity and mislocalization of membrane proteins (shown by mislocalization of Crumbs, DE-Cadherin, and Delta). As a consequence, *aqz* mutant embryos with compromised apical-basal cell polarity develop spotty changes of neuronal and epithelial numbers of cells.

## INTRODUCTION

Cell polarity is a crucial crossroads for epithelial development and function. Epithelial polarization generates and maintains differentiation of apical and basolateral membrane domains. In fruit fly embryos, cellularization and cell polarity come first and preempt gastrulation and further development (*Campos-Ortega & Hartenstein, 1997*). Cellular blastoderm fly embryos exhibit cell polarization (*Nusslein-Volhard & Wieschaus, 1980*). For this, both membrane-attached and cytosolic cytoskeletal proteins are required. Failure of proper polarization leads to early death with a very early phenotype during cellularization and/or germband extension (*Pai et al., 1996*; *Peifer & Wieschaus, 1990*; *Tepass et al., 1996*).

Once attained, cellular polarization must be regulated and maintained (*Campos-Ortega & Hartenstein, 1997*). A well-studied case is neurulation in the fly. The embryonic tissue that gives rise to both fly neural and skin precursors is the ventral ectoderm, or neuroectoderm (*Campos-Ortega & Hartenstein, 1997*; *Campos-Ortega & Knust, 1990*).

Corresponding author
Juan R. Riesgo-Escovar,
juanriesgo@prodigy.net.mx

Early positional cues establish the neuroectodermal region in the procephalon and the ventrolateral ectoderm (*Egger, Chell & Brand, 2008*; *Technau, Berger & Urbach, 2006*). Within this epithelial sheet, initially equivalent groups of cells, the proneural groups, form, evidenced by expression of marker genes, like the basic-helix-loop-helix (bHLH) *achaete-scute* complex and molecules of the Notch (N) signaling pathway. Among these cells, Notch signaling limits the number of neural precursors that delaminate and form neuroblasts (*Bertrand, Castro & Guillemot, 2002*; *Campos-Ortega & Knust, 1990*; *Cornell & Ohlen, 2000*; *Hartenstein et al., 1992*; *Portin, 2002*; *Skeath et al., 1992*; *Technau, Berger & Urbach, 2006*; *Zhao, Wheeler & Skeath, 2007*). From each individual proneural cluster, normally just one cell differentiates as a neuroblast (*Cabrera, 1990*; *Fortini, 2009*; *Fortini & Artavanis-Tsakonas, 1993*; *Schrons, Knust & Campos-Ortega, 1992*). Neuroblasts migrate and divide to give rise to ganglion mother cells, and ultimately, to neurons and glia (*Karcavich, 2005*; *Zhong & Chia, 2008*). Non-selected cells receiving the Delta signal remain in the ectoderm and become epidermoblasts, differentiating into skin cells (*Hartenstein & Wodarz, 2013*; *Portin, 2002*). The process necessitates maintenance of polarity and changes in cell shape (*Harris & Tepass, 2008*; *Harris, 2012*; *Wang et al., 2004*).

Recent studies show that cell polarity regulatory proteins (CPRPs), such as Cdc42, Par complex components, and WASP proteins establish apical-basal cell polarity (*Bayraktar, Zygmunt & Carthew, 2006*; *Harris & Tepass, 2008*; *Krahn & Wodarz, 2009*; *Tepass, 2012*). Epithelial defects like loss of cell dissociation and apical-basal polarization affect adherens junctions (AJ), and the way in which Dl and N interact (*Buszczak et al., 2007*). Membrane polarization and Dl activation are critical events during early neurogenesis (*Bayraktar, Zygmunt & Carthew, 2006*; *Fehon et al., 1991*; *Harris & Tepass, 2010*; *Kooh, Fehon & Muskavitch, 1993*; *Liu et al., 2012*; *Weinmaster & Fischer, 2011*). Disrupted CPRPs alter epithelial integrity and cause abnormalities in neuroblast proliferation and morphogenetic movements during early neurogenesis. This may result in nervous system hyperplasia paralleled by loss of epidermis, a typical "neurogenic" phenotype (*Bayraktar, Zygmunt & Carthew, 2006*; *Harris & Tepass, 2008*; *Harris & Tepass, 2010*; *Krahn & Wodarz, 2009*; *Wang et al., 2004*; *Weinmaster & Fischer, 2011*).

Here we characterize the *aaquetzalli* (*aqz*) gene, which encodes several alternatively spliced transcripts expressed throughout the life cycle. *aqz*, meaning 'fan' in nahuatl, and the name refers to the abnormal embryonic cuticular phenotypes of mutant embryos. Mutations in *aqz* are lethal, with an extended phenocritical period. Previous studies suggest that *aqz* is required pleiotropically during *Drosophila* embryogenesis: *aqz* mutant germline clones die with cuticular holes (*Perrimon et al., 1996*), and *aqz* RNAi injected in developing embryos show axon guidance and synaptogenesis defects (*Ivanov et al., 2004*). These mutant phenotypes are in ectodermally derived cells (epidermis and nervous tissue), despite affectation in all cells in the first case, and generalized injection of RNAi in developing embryos in the second case. Therefore, these experiments support the tenet that *aqz* is required in ectodermal tissues.

In order to fully characterize the locus, we generated and identified new mutant *aqz* alleles. These alleles show new, more generalized zygotic neural and epidermal mutant phenotypes similar to N pathway mutants ("neurogenic" phenotypes). *aqz* is expressed

early in the ectoderm and ectodermally derived tissues. Our findings describe and establish an early role for *aqz* in ectoderm differentiation as a modulator of cell polarity, promoting apical polarization to maintain epithelial integrity. *aqz* modulates Dl membrane localization during neural lateral inhibition.

## MATERIALS & METHODS

### Fly strains, genetics and husbandry

Flies were housed in standard conditions (25 °C, 12:12 light:dark cycles and 50% humidity) in standard food vials (unrefined sugar-baker's yeast-agar-gelatin-water). Oregon R and *yw* were used as control stocks. $aqz^2$, $aqz^6$, and $aqz^7$ are lethal insertion alleles. $aqz^2$ is Bloomington stock #14691 ($y^1$ $w^{67c23}$($y^1$ $w^{67c23}$; $P\{y[+mDint2]$ $w[BR.E.BR]=SUPor-P\}KG08159$ $ry^{506}$); $aqz^6$ is Bloomington stock # 20315 ($y1$ $w67c23$; $P\{EPgy2\}CG9821EY11495$). $aqz^7$ was a lethal insertion line that is now lost (former Bloomington stock #1492). $aqz^{GFP}$ is also a transposon insertion, and was stock #CB03335 from the former FlyTrap Stock Collection at Yale University, USA. It is a lethal insertion that generates chimeric proteins GFP::Aqz. New *aqz* alleles generated: $aqz^1$, $aqz^3$ and $aqz^5$, are excision events. $aqz^3$ is an excision event from the now lost $aqz^7$ insertion. $aqz^1$ and $aqz^5$ are excisions from $aqz^2$. All excision alleles are lethal.

$aqz^{rescue}$ was generated by transgenesis using P[acman] BAC #CH322-25015 (*Venken et al., 2009*). CH322-25015 has approximately a 19 Kb fly genomic fragment from 4631615 to 4650999 on chromosome three, including the complete *aqz* locus (CG9821), CG9836 or *IscU*, *iron sulfur cluster assembly enzyme* (*Tian et al., 2014*), where the known mutants alleles die between late larval life and pupariation, CG8369 (an undescribed Kazal domain containing protein), CG9837 and CG8359, or *hng2*, *hinge2*, plus a lncRNA: CR43130, that partially overlaps CG9821, but in the opposite orientation. The whole genomic rescue construct was inserted on 2R by recombineering at 53B2 (using Bloomington line #9736). We also used three different deficiencies that uncover the *aqz* locus to perform complementation tests: Bloomington stock # 7629 ($w^{1118}$; *Df(3R)Exel6150*, $P\{XP-U\}Exel6150/TM6B$, $Tb^1$); Bloomington stock # 24982 ($w^{1118}$; *Df(3R)BSC478/TM6C*, $Sb^1$ $cu^1$), and Bloomington stock #25010 ($w^{1118}$; *Df(3R)BSC506/TM6C*, $Sb^1$ $cu^1$).

### *in situ* hybridization

Embryo *in situ* hybridization was done following (*Tautz & Pfeifle, 1989*), with few modifications for fluorescence detection. Briefly, whole embryos were fixed with formaldehyde, and hybridized with digoxigenin RNA probes overnight at 55 °C in hybridization buffer. *aqz* 5′ RNA probes (derived from clone LD47990, which corresponds to the 5′ end of *aqz*, sense and anti-sense) were synthesized using Roche's digoxigenin RNA labeling kit following the manufacturer's protocol (Roche, Basel, Switzerland). The fixed and hybridized embryos were incubated with polymerized HRP-conjugated anti-digoxigenin antibody at a 1:2000 dilution for 1 hr 45 min at room temperature, washed and incubated with 1:10 diluted Tiramidine-CY3 for 45 min, using the tyramidine derivative system (TSA; Perkin Elmer, Waltham, MA, USA), and mounted in Vectashield (Vector Laboratories, Burlingame, CA, USA).

## Northern blot analysis

Total RNA was isolated from 8–12 hr old embryos, 2nd and 3rd instars larvae, pupae, and adults with Trizol, according to the manufacturer's instructions (Gibco/BRL, Waltham, MA, USA). Northern blotting was done according to standard procedures (*Sambrook, Fritsch & Maniatis, 1989*). Approximately 3–4 micrograms of total RNA was loaded per lane. DNA probes were from cDNA clone LD47990 (*Stapleton et al., 2002*), for the *aqz* 5′ region, which is common to all *aqz* transcripts, and clone LD02090, which labels the 3′ end of the gene not present in *aqz* RA, but present in *aqz* RB, RC, and RD. The probes were labeled with a-$^{32}$P deoxycytidine 5′-triphoshpate, following *Feinberg & Vogelstein (1984)*. Both cDNA clones were purchased from the Berkeley Drosophila Genome Project clone collection.

## Antibody generation

A rabbit polyclonal antiserum was generated against the KSIRLKKGAC peptide, representing the Aqz amino terminus. Immunoglobulins G (IgGs) were affinity purified from immune serum (New England Peptide, MA, USA). This antibody failed to detect Aqz proteins in Western blots, but detectd Aqz in tissue sections.

## Western blot

Embryos were homogenized and immunoblotted following (*Wodarz, 2008*). Proteins were separated on 10% denaturing acrylamide gels (SDS-PAGE), transferred to nitrocellulose membranes and incubated with rabbit anti-GFP (1:5000, SC-8334, Santa Cruz Biotechnology, USA) as primary antibody. We pre-absorbed anti-GFP before use by incubating diluted primary antibody with wild type fixed embryos in incubation solution overnight. We used Alkaline Phosphatase (AP)-conjugated secondary antibodies (goat anti-rabbit 1:1000 from Zymed; Zymed, South San Francisco, CA, USA). Secondary antibody was incubated for 1 h at room temperature. We used the NBT/BCIP substrate solution (Roche Diagnostics, Basel, Switzerland) for AP detection.

## Immunohistochemistry

We followed the embryo staining protocol in (*Karr & Alberts, 1986*), except for *aqz*$^{GFP}$ embryos. The *aqz*$^{GFP}$ embryos were washed in methanol only once, and rehydrated immediately. For primary antibodies, we used rat anti-Deadpan (1:2, gift of Cheng-Yu Lee), rabbit polyclonal anti-Aqz (1:500 this work), and rabbit anti-GFP (1:100, SC-8334, Santa Cruz Biotechnology, pre-absorbed with wild type embryos in incubation solution overnight before use). Polyclonal rat anti-Elav (7E8A10, 1:250), polyclonal rat anti-DE-cad (DCAD2, 1:10), mouse monoclonal anti-FASIII (7G10, 1:100), mouse monoclonal anti-EVE (3C10, 1:10), mouse monoclonal anti-Coracle (C615.16, 1:250), mouse monoclonal anti-22C10 (1:200), mouse monoclonal anti-BP102 (1:200), mouse monoclonal anti-Repo (8D12, 1:200), mouse monoclonal anti-Dl extracellular domain (C594.9B, 1:100), mouse monoclonal anti-Crumbs (Cq4, 1:100), and mouse monoclonal anti-Dlg (4F3, 1:3) were from the Developmental Studies Hybridoma Bank, University of Iowa, Iowa City, IA, USA. Primaries were incubated with fixed embryos overnight at 4 °C. Embryos were then washed 4 times for 20 min each using 0.3% Triton X-100 in PBS. Anti-Deadpan,

anti-Aqz, anti-Eve, anti-DE-cadherin, anti-GFP and anti-Delta antibody signals were amplified using biotin-conjugated secondary antibodies in combination with the Vector Elite ABC kit (Vector Laboratories, Burlingame, CA, USA) according to the manufacturer's instructions. Fluorescent-coupled (1:1000, Zymed, South San Francisco, CA, USA) and biotin-conjugated secondary antibodies (1:500; Santa Cruz Biotechnology, Santa Cruz, CA, USA) were incubated for 2 h at room temperature. For embryos incubated with biotin-conjugated antibodies, we used the dye-labeled tyramidine derivative system (TSA) for signal amplification. After washing 4 times with 0.3% Triton X-100 in PBS, fluorescently labeled embryos were mounted in Vectashield.

## Cuticle preparations

We followed the protocol described by *Nusslein-Volhard & Wieschaus (1980)*. Embryos and 1st instar larvae were collected 24 h after egg laying, dechorionated in 50% bleach for 5 min, rinsed in distilled water and placed in an Eppendorf tube containing 50% heptane and 50% methanol, and shaken vigorously for 2 min. The upper phase was removed and both embryos and 1st instar larvae were washed with 100% methanol three times. Embryos and 1st instar larvae were rehydrated with PBT (PBS + 0.3% Triton X-100). Once rehydrated, they were carefully dropped onto a glass slide and covered with mounting medium (Hoyer's Medium or PVA) and coverslipped. They were incubated at least one day in a hot plate. Cuticles were visualized using dark field microscopy.

## Embryonic lethality

To assess embryonic lethality, we separated and collected $aqz^{GFP}$ homozygote eggs from 24 h egg-laying plates under a dissecting microscope equipped with a blue light source and a GFP filter. Homozygote mutant embryos fluoresce with GFP, and can be distinguished from control (non-fluorescent) or heterozygous (weakly fluorescent) eggs. These were placed on fresh egg-laying plates alongside wild type, control Oregon R eggs from the same day, and incubated a further 72 h. At the end, non-hatching, dead embryos were counted.

## DNA sequencing

Automated sequencing was performed using an ABI 310 sequencer. We sequenced fully cDNAs SD19655, LD74990, RE74095, LD18978 and LD02060. For $aqz^1$, we selected homozygous mutant embryos by lack of GFP expression present in the balancer chromosome, and made a crude homogenate from them, to use for PCR amplification. We PCR-amplified using primers AqzA (TACACCACCGTCTATTGCG) and AqzB (CCCCATTGTGTCAGATTTCTT), amplifying the *aqz* promoter region for $aqz^1$. We cloned the ensuing PCR products in pGEMT Easy vector (Promega, Madison, WI, USA). The PCR product inserts were sequenced fully. We sequenced three independent mutant clones, and all confirmed the mutation in the promoter region of $aqz^1$.

## Genetic assays

Crosses or stocks were let to lay eggs in agar plates, and eggs laid were collected, classified, and counted. We selected stage 14–15 embryos (late neurogenesis). Embryos were fixed and stained using anti-Elav (NS) and anti-Coracle (epidermis), to score neurogenic

phenotypes. The percentage of embryos with wild type (WT, no defects in the NS and epidermis) or "neurogenic" phenotype was calculated by using the following formula: % of WT or neurogenic embryos = WT or neurogenic embryos/total of homozygotes embryos counted*100. Fisher's exact test was used to assess significance of the differences indicated. We scored at least 400 embryos for every experiment.

### *aqz* rescue experiments

To rescue *aqz* in an endogenous genomic context that includes the whole locus and alternatively spliced transcripts, we used a genomic BAC (#CH322-25015) that has a genomic insert of 19, 384 bp in the attB-P[acman]-CmR-BW vector, roughly centered on CG9821 encompassing the whole locus, but including four other neighboring genes (CG9836, CG8369, CG9837, and CG8359, plus a lncRNA: CR43130, that partially overlaps the *aqz* sequence in the opposite orientation). These other loci, excepted *aqz*, have no known lethal alleles. Of these, only *IscU* (CG9836) and *hng2* (CG8359), besides *aqz*, are expressed significantly during embryogenesis (there is mid- to late embryonic expression of CG9837) (*Dos Santos et al., 2015*). Mutant alleles or insertions in these loci (stocks # 22145, 34193, 31835, 14691) behave differently from $aqz^{GFP}$, as they fully complement the *Df(3R)Exel6150* that is embryonic lethal over $aqz^{GFP}$, and not surprisingly fully complement $aqz^{GFP}$ as well (Table S1). We inserted this genomic construct, thereafter called $aqz^{RESCUE}$ construct, by standard methods at an attB acceptor site at 53B2, as stated above. This insertion has no phenotype in a wild type background. We then constructed stocks $aqz^{RESCUE}$; $aqz^{mutant}$ and controls that were doubly balanced with the T(2:3)*CyO-TM3,Ser,GFP* fused chromosome two and three balancers (Bloomington stock #5703). Homozygous mutants either with one or two copies of the $aqz^{RESCUE}$ construct were assessed for rescue both by staining embryos as above for neurogenic phenotypes, and for adult viability. For adult viability, homozygous mutant embryos and controls with and without rescue construct present were collected from egg lays, allowed to develop, and emerging adults were scored. Fisher's exact test was used to assess significance of the differences.

## Image acquisition and processing

We studied embryos using light, fluorescence, and laser scanning confocal microscopy (LSM). Cuticles and immunofluorescence of whole embryos were imaged using a digital camera (CoolSnap; Photometrics, Tuscon, AZ, USA) on a Nikon E600 Eclipse microscope with Plan-Fluor 20X/0.50 NA and a Nikon Super High Pressure Mercury Lamp. Confocal sections of embryos were made in a Zeiss LSM 510 Meta with Plan-Neofluar 25x/0.8 NA and a Plan-FLUAR 100X/1.45 NA objectives (Carl Zeiss, Oberkochen, Germany), or a Zeiss LSM 780 with Plan-APOCHROMAT 25X/0.8 NA and Plan-APOCHROMAT 63X/1.40 NA objectives (Carl Zeiss, Oberkochen, Germany) at 18 °C. For cuticle and fluorescence image processing, we use iVision 4.0.12 software (BioVision Technologies, Milpitas, CA, USA). Confocal sections were processed using the Image Browser software (Carl Zeiss, Oberkochen, Germany) used to generate image stacks. All captured images were processed using Adobe Photoshop software.

### Ribosome footprinting, mRNA-seq, sequence visualization and alignment

We used raw data generated and publish by (*Dunn et al., 2013*) available in NCBI's Gene Expression Omnibus (*Edgar, Domrachev & Lash, 2002*) under GEO series accession number GSE49197. We used the Integrated Genome Browser (IGB) software for visualization and exploration of the *aqz* genome locus and corresponding annotations (*Nicol et al., 2009*).

## RESULTS

### The *aqz* locus

The isolated *aqz* transposon insertions map to a locus (CG9821 in Flybase) at 85B2 on the right arm of the third chromosome. This locus has alternative splicing at the end of the first exon and different 3′ ends, as judged from ESTs from the locus, and gives rise to at least three different transcripts differing in size and composition (Figs. 1A, 1C (*Dos Santos et al., 2015*)). The CG9821 locus spans over 7Kb, partially overlapping a non-coding RNA gene at its 5′ end (CR43130). It also overlaps CG9837 at its 3′ end (at least 89 bp of the 3′ end of CG9821 overlap the CG9837 5′ end). CG9837 is transcribed in the same direction as CG9821. The 5′ end of CG9821 has a proline-rich domain found in nuclear receptor co-activators (PNRC-domain proteins), present in invertebrate and vertebrate proteins (Fig. 1B).

### *aqz* cDNAs and proteins

We identified and sequenced cDNAs and genomic fragments from the *aqz* locus. There is a wealth of ESTs isolated from the CG9821 region, consistent with widespread expression. We sequenced fully several of these ESTs and pieced them together with other sequences available. The two exons composing the transcribed part of the locus vary in size, due to alternative splicing in the first exon and different endings of the second exon (Fig. 1A).

The Aqz protein harbor a domain with significant homology to a proline-rich conserved domain of nuclear receptor co-activators (PNRC-domain proteins; Fig. 1B). This conserved domain, a proline rich, SH3 protein-protein interaction domain thought to mediate union to several nuclear receptors in vertebrates, is present in both vertebrate and invertebrate proteins, and is 60 aa long. Within it, a YAG motif besides several proline residues and two more carboxy-terminus-located small regions of homology constitute the conserved core of the domain. Vertebrate proteins are thought to interact with nonsense-mediated RNA decay proteins (*Lai et al., 2012*), as well as with a variety of nuclear receptors. Knock out mice for PNRC2 are viable and fertile, but males exhibit a lean phenotype due to higher energy expenditure (*Zhou et al., 2008*). Whether fly proteins also exhibit similar binding and functional roles is as yet unclear. *Drosophila melanogaster*, among sequenced Drosophila species, possesses a second uncharacterized protein with this motif: CG32797 (Fig. 1B), mutations in which may have similar phenotypes to those encountered in mice PNCR2 knockouts. It remains to be seen whether any Aqz mutant has higher energy expenditure, or similar phenotypes to the PNCR2 knockouts.

The theoretically reconstructed cDNAs are somewhat at variance from data published for CG9821 and CG9837 (*St Pierre et al., 2014*). cDNAs predicted for CG9821 are shorter

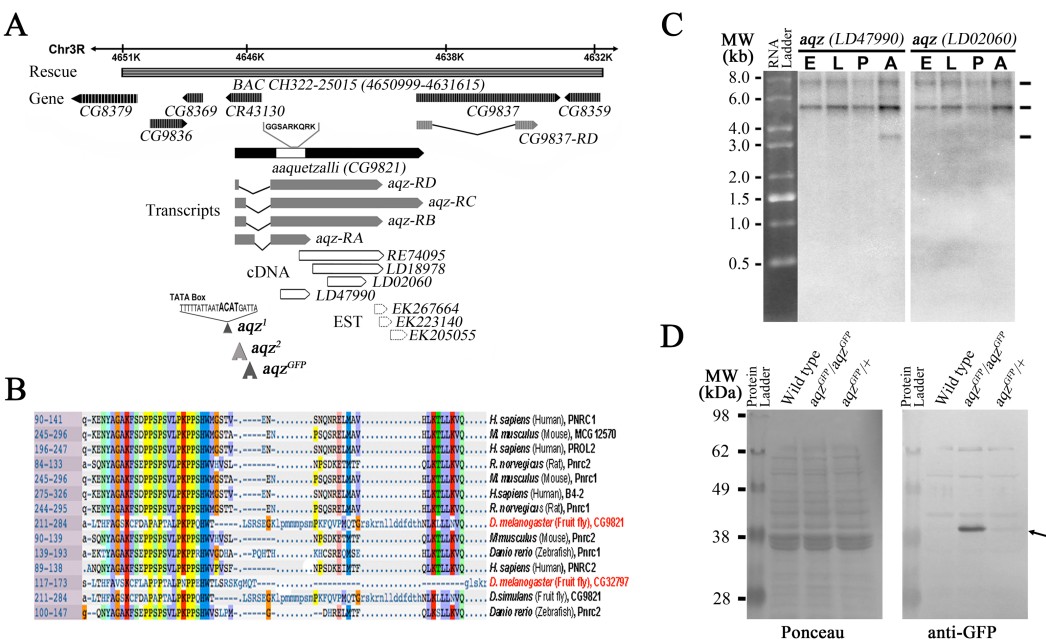

**Figure 1** **The *aaquetzalli* (*aqz*) locus codes for at least three alternatively spliced transcripts.** (A) *aqz* encompasses a little above 7 Kb, from 4647000 to 4638000 at the base of chromosome 3R at 85B2; in the figure the genes are oriented such that the 5′ end of *aqz* is to the left. Besides *aqz* (solid black arrow), other genes mapping in the region are shown as grey-hashed arrows. *aqz* codes for four predicted different mRNA species, each with two exons (grey boxes) and one intron (black line). Several cDNAs and ESTs (white boxes with black borders, and white boxes with black dotted lines, respectively) map in the locus. Underneath the depicted chromosome line a grey bar signals the extent of the genomic rescue construct employed (BAC CH322-25015), inserted in the second chromosome at 53B2 (Bloomington line # 9736). The location of two transposon insertion mutants, one in the promoter region (*aqz²*), and one in the intron (*aqz^GFP*), are shown. *aqz^GFP* is a fly trap line (#CB03335), coding for a predicted fusion protein, where the GFP moiety is fused to the 5′ end of Aqz. *aqz¹* has a four base pairs insertion, depicted in bold, in the promoter region of *aqz* between the predicted TATA box and initiation of transcription. *aqz¹* was isolated as an imprecise excision of the *aqz²* transposon. Nearly all isolated *aqz* alleles are lethal. (B) Aqz has a conserved domain at the 5′ end of the protein. The homology is for a proline-rich conserved domain of nuclear receptor co-activators (PNRC-domain proteins) in single letter amino acid code, present in both invertebrate and vertebrate proteins. Numbers to the left indicate position of the domain within the proteins. (C) Northern blot using two different *aqz* cDNAs as probes reveal three different transcripts of 7.53, 5.0 3 and 3.16 Kb. The smallest band (3.16 Kb) is only present in adult flies, and is only seen using a 5′ cDNA (LD47990), present in all *aqz* isoforms. The same blot that was used with the first probe, was photographed, then stripped, and re-probed with the a more 3′ cDNA (LD02060), that recognizes *aqz* RB, RC, and RD, but not *aqz* RA. $n > 10$. (D) Aqz mutant and control Western blot using an antibody against GFP recognizes a band at approximately 40 kD (right black arrow) in *aqz^GFP* homozygotes, and weakly, in *aqz^GFP* heterozygotes. All other weak bands are present in all lanes, including the control, wild type lane, and are, consequently, not specific.

than the ones we pieced together, but the annotations do not take into consideration several ESTs located towards the 3′ end of CG9821, also reported, that partially overlap other identified ESTs for CG9821 and CG9837 (Fig. 1A; Fig. S1), and the RNAseq data posted from the modEncode project, which shows that the two annotated loci overlap (*Celniker et al., 2009*). There are four predicted different isoforms for CG9821 mRNAs, all with the same coding sequence, two of which are very similar in size (*aqz-RD* and
*aqz-RB*), varying only in the size of the first exon (cDNAs of 4950 and 4719 nucleotides long, respectively). The size of the shortest predicted cDNA isoform (*aqz-RA*) is just 2357 nucleotides long. The longest predicted cDNA (*aqz-RC*) is approximately 6145 nucleotides long and it overlaps at least 89 bp with the CG9827 locus (Fig. 1A).

In order to contrast these reconstructions and annotations from ESTs and gene-prediction software (*Celniker et al., 2009*), we performed Northern blots. We identified in Northern blots with 5′ probes at least three differently sized mRNAs that correspond roughly in size to the ones we reconstructed, two of which are expressed throughout the fly life cycle (Fig. 1C). This last agrees well with published data showing *aqz* expression at all life cycle stages and in multiple tissues including the brain, ovary, imaginal discs, and larval salivary glands (*Jagadeeshan & Singh, 2005*; *St Pierre et al., 2014*).

We observe a darker band at slightly over 5 Kb that may correspond to the two 4.9 and 4.7 bands annotated for CG9821. We observe a fainter larger band at 7.5 Kb; this last may correspond to the longest reconstructed cDNA spanning CG9821 and overlapping CG9837. We also observe a smaller band at 3.1 Kb with a 5′ probe; the closest annotated CG9821 mRNA is 2.35 Kb long. This smaller mRNA species is only seen in adult flies with a 5′ probe, and not with a more 3′ probe, showing that the small cDNA comes from the 5′ end of the locus, as predicted for the 2.35 Kb species. These cDNAs share the same 5′ end harboring the PNRC domain, but have different 3′ extensions (Fig. 1C). These extensions are mostly predicted as non-coding (*St Pierre et al., 2014*). Since the 5′ coding sequence is common for the cDNAs, we generated an antibody against an Aqz amino terminal peptide fragment. Unfortunately, we were not able to obtain a clear and specific signal using this antibody in Western blots.

We also analyzed ribosome occupancy data published for the locus, using data from a modified protocol of ribosome profiling assay. This assay has helped reveal that stop codon read-through plays an important regulatory role in N-terminally and C-terminally protein extension during mRNA translation (*Dunn et al., 2013*). We performed an *aqz* mRNAs ribosome footprint analysis using RNA abundance and ribosome density data generated and published (*Dunn et al., 2013*), available at NCBI GEO (http://www.ncbi.nlm.nih.gov/geo/), and viewed the display alignments using the Integrated Genome Browser 8.1 (IGB) sequence viewer software (Fig. 2). The analysis reveals that the first exon is transcribed as well as infrequent 3′ extensions in the second exon, beyond the predicted stop codon. These data are consistent with our Northern results (Fig. 1C).

### *aqz* alleles

We identified and generated new *aqz* alleles. Two previously described mutant alleles appear to be weak loss-of-function alleles (named here $aqz^2$ (also known as P14691), an insertion in the promoter, and $aqz^7$ (also known as P1492), an insertion in an intron, now lost *Perrimon et al., 1996*; Fig. 1A). We generated imprecise excision lines. Mutant embryos have zygotic embryonic mutant phenotypes affecting the central NS and the cuticle (Fig. S2). The RNAi study referred to above also pointed to an *aqz* requirement in axon guidance and synaptogenesis (*Ivanov et al., 2004*).

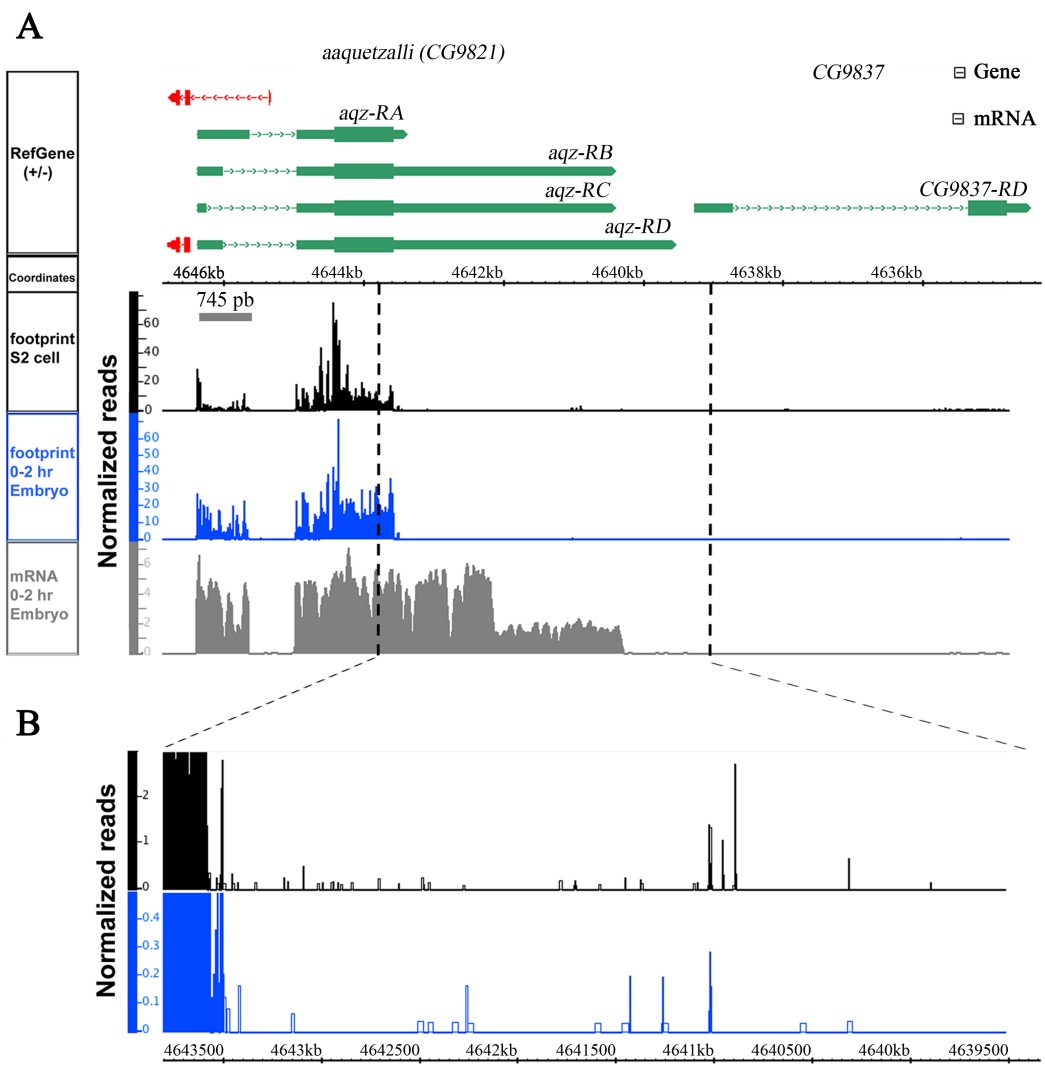

**Figure 2** *aqz* **has alterative splicing.** *aqz* mRNAs first exon is translated, and some have also 3′ stop codon read-through in the second exon, judging from ribosome footprinting data. The *aqz* locus (CG9821) together with the annotated predicted mRNAs are depicted on top (green) with dotted lines depicting introns, and red lines depicting other transcripts. The neighboring CG9837 locus, overlapping CG9821 at the 5′ end, is also depicted in green. The ribosome footprinting data is depicted below taken from S2 cells (black), 0–2 hrs. embryos (blue) and mRNA-seq (gray) in reads per million. Inset shows a magnification of the Aqz C-terminal region where protein extension may occur.

The new alleles generated by imprecise excisions are lethal, and from these we characterized $aqz^1$ further. Molecular analysis of $aqz^1$ revealed a 4 bp insertion in the promoter region of CG9821 (sequence inserted: ACAT, 189 bp 5′ from the transcription start site, and 171 bp 5′ from the parental P transposon insertion), probably due to the imprecise *P*-element excision event (Fig. 1A). This 4 bp insertion is not present in control strains, in other *aqz* mutants, or in revertants from the same *P*-element excision experiment.

**Table 1** *aqz* **alleles complementation matrix.** *aqz^insertion* (*aqz²*) is homozygous viable and complements *aqz¹*, which is homozygous lethal. *aqz^GFP* (another *P*-element insertion), is homozygous lethal. *aqz¹* complements *aqz^GFP* and all three *aqz*-uncovering deficiencies (*Df(3R)Exel6150*, *Df(3R)BSC478*, and *Df(3R)BSC506*). *aqz^GFP* does not complement these deficiencies. + Indicates complementation, L indicates non-complementation (lethality), HV indicates homozygous viable, and HL indicates homozygous lethal. *Df(3R)Exel6150* deletes 85A5-B6, *Df(3R)BSC478* deletes 85A5-B8, *Df(3R)BSC506* deletes 85B1-C2. Together, the common deleted fragment goes from 85B1-B6. The *aqz* locus is located at 85B2.

| Allele | *aqz¹* | *aqz²* | *aqz^GFP* | Df(3R)Exel6150 | Df(3R)BSC478 | Df(3R)BSC506 |
|---|---|---|---|---|---|---|
| *aqz¹* | HL | + | + | + | + | + |
| *aqz²* | | HV | + | + | + | + |
| *aqz^GFP* | | | HL | L | L | L |
| Df(3R)Exel6150 | | | | HL | L | L |
| Df(3R)BSC478 | | | | | HL | L |
| Df(3R)BSC506 | | | | | | HL |

We also identified another lethal allele, *aqz^GFP*, from a collection of gene traps in the former Yale Gene Trap stock center (*Kelso et al., 2004*). This allele, *aqz^GFP*, has an insertion in the first intron/first exon (depending on the splice variant), coding for GFP, theoretically giving rise to a chimeric protein made of GFP fused to the Aqz amino terminus (GFP::Aqz). As shown in Fig. 1D, we detected only one band of approximately 40 kDa in Western blots from *aqz^GFP* homozygous mutant embryos using an anti-GFP antibody. This band must contain GFP (of approximately 26.9 kDa), and a fragment of the Aqz protein product (of approximately 13 kDa).

In order to genetically characterize the *aqz* locus, we conducted complementation tests of lethal mutant alleles. The *aqz¹* and *aqz²* alleles complement each other and three different deficiencies uncovering the locus. These mutant alleles also complement the *aqz^GFP* allele. In contrast, *aqz^GFP* does not complement the same three different deficiencies that uncover the locus, effectively mapping *aqz^GFP* to the common interval uncovered by them at 85B, where the *aqz* locus, and the insertion points of *aqz^GFP* and the other *aqz^insertions*, lie (for example, *aqz²* and *aqz^GFP* lie only 322 bp apart within the promoter/first exon/intron of the locus), yet they complement (Fig. 1A; Table 1). The fact that *aqz¹* complements the three deficiencies used (that do not complement *aqz^GFP*) and *aqz^GFP* could also be explained saying that the lethality of *aqz¹* maps elsewhere, despite the molecular lesion and similar mutant phenotypes. All *aqz* mutants, despite their origin, have the same mutant phenotypes. In all, a fraction of mutant embryos have central NS defects and holes in the cuticle (Figs. 3A–3F, Figs. S2 and S3A–S3F). As *aqz^GFP* behaves genetically as a loss-of-function allele (does not complement all the deficiencies uncovering the locus tested with similar phenotypes and theoretically all Aqz protein isoforms are affected), we mainly focused our studies on *aqz^GFP*.

We used a genomic construct encompassing the *aqz* locus, and performed rescue experiments for *aqz^GFP* and *aqz¹* (Figs. 1A, 3G–3J–Figs. S3G–S3I). Rescue experiments show that *aqz^GFP* behaves as a loss-of-function allele, as a significant proportion of mutant embryos survive embryogenesis and larval stages, and go on to emerge as viable adults

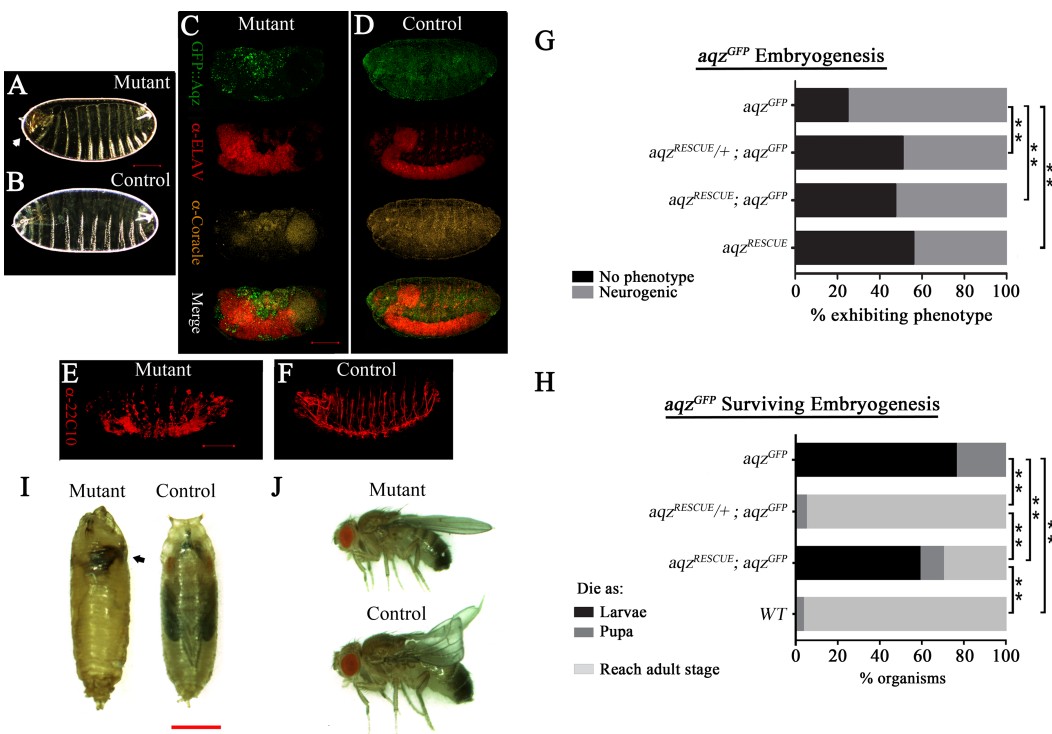

**Figure 3** *aqz^{GFP}* **mutant phenotypes.** (A) and (B) show mutant (*aqz^{GFP}*/*aqz^{GFP}*) and wild type cuticles, respectively, the former one with an anterior hole and head involution defects (white arrow). (C) and (E) show mutant embryos stained with neuronal and epithelial markers: in (C) anti-Elav for neuronal bodies and anti-Coracle (Cora) for epithelial cells, while in (E) 22C10 for axons and some neuronal bodies. (D) and (F) show the same stainings using wild type control embryos. Scale bar is 100 μm. (G) Shows the percentages of embryos with neural phenotypes in *aqz^{GFP}*/*aqz^{GFP}* embryos, with one or two copies of a genomic rescue construct. One copy of the rescue construct rescues significantly the mutant phenotype, whereas two copies effects a significant, but smaller, rescue, pointing to some gain-of-function effects with two rescue constructs. Two copies of the rescue construct in a wild type background have neurogenic phenotypes in over 40% of embryos, phenotypes similar to *aqz* mutants. In (G) and (H) *aqz^{GFP}* is homozygous mutant, *aqz^{RECUE/+}* is one copy of the rescue construct, and *aqz^{recue}* indicates two copies of the rescue construct. (H) Extended phenocritical period of *aqz^{GFP}*/*aqz^{GFP}* mutants. The surviving *aqz^{GFP}*/*aqz^{GFP}* mutant embryos die majorly as larvae, with approximately 20% reaching pupariation. No adults are ever observed. The *aqz^{GFP}*/*aqz^{GFP}* lethality is rescued to wild type levels with one copy of the *aqz* genomic rescue construct, but two copies of the rescue construct, while having a significant rescue effect, are less effective, pointing to some gain-of-function effects with two rescue constructs. (I) *aqz^{GFP}*/*aqz^{GFP}* mutant pupae die early, with necrosis (black arrow), compared to a control pupae. Scale bar is 1 mm. (J) Rescued *aqz^{GFP}*/*aqz^{GFP}* mutant fly, compared to sibling control fly (has curved wings as marker). In (A–H), ** denotes significant differences. In (A–H), for all embryo experiments, representative ones are shown, and n examined is over 200 per condition.

(fertile males and unfertile females) using one copy of the rescue construct (Figs. 3H–3J). All the foregoing data, taken as a whole and in conjunction with published data, establishes the CG9821 transcription unit as being the *aqz* locus.

## Gain of function phenotypes

Having two copies of the rescue construct did not increase embryonic rescue; rather, it proved deleterious for larvae, as many died as larvae, and, as a consequence, significantly

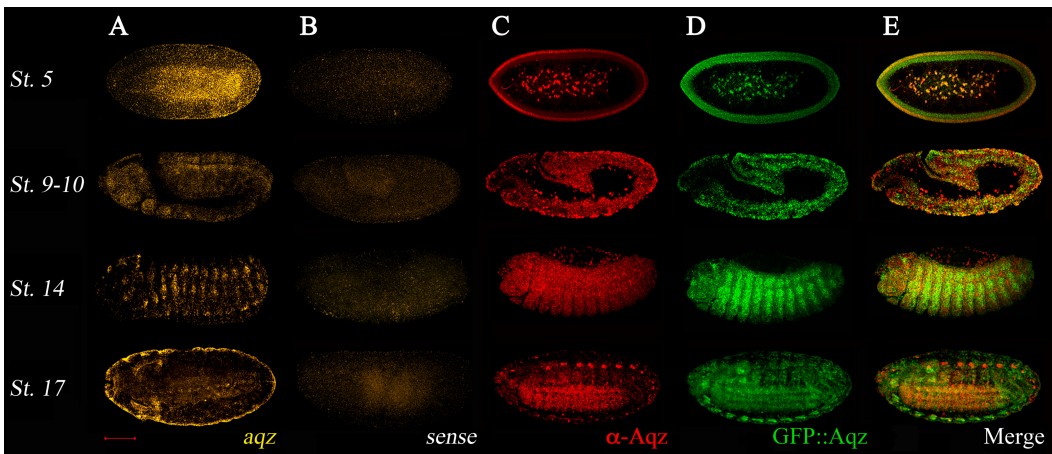

**Figure 4** **Aqz mRNAs and proteins are expressed throughout embryogenesis in the ectoderm and in ectodermally derived structures, and have maternal contribution.** (A) Anti-sense *aqz in situ* hybridization results at different embryonic stages of development; (B) shows results with the sense control probe (both derived from *aqz* cDNA LD47990). Embryos in (C) are stained with the anti-Aqz antibody, (1:100), whereas (D) shows expression of chimeric GFP::Aqz proteins using an anti GFP antibody (1:100) from the same embryos, which are heterozygous for $aqz^{GFP}$, and thus have expression of both GFP::Aqz and native Aqz protein. As shown in (E), results from both are largely coincidental except some PNS cells and some amnioserosa cells, only labeled with the anti-Aqz antibody. In (A–E), anterior is left and dorsal is up. All views are lateral views except the lower row, where ventrolateral views are shown to evidence the developing central nervous system. Stage 5 early syncytial embryo, stage 9–10 germband extension/retraction stages, stage 14 dorsal closure, and stage 17, end of embryogenesis. Scale bar is 100 μm.

fewer reach pupation, and adulthood (Fig. 3H). Two copies of the rescue construct in a wild type background yielded a fraction of embryos with neurogenic defects (Fig. 3G), showing that loss and gain of function conditions give similar results. These results suggest that Aqz isoforms abundance and proportions are critical for normal development. Consistently, $aqz^1$ mutants show no adult rescue; rather, if done with one or two rescue constructs, results show a significant increment in the frequency of mutant phenotypes. This is consistent with at least a partial gain-of-function nature for $aqz^1$, as the small Aqz protein is still present in $aqz^1$ (Figs. S3G–S3H). Together with the rescue results of $aqz^{GFP}$ and overexpression, this leads support to the notion that the ratio and abundance of different Aqz protein isoforms is critical. We focused our studies on the embryonic stages, where defects are first observed.

### *aqz* embryonic expression

*aqz* embryonic expression is dynamic (Fig. 4). Expression is largely coincidental using a 5′ *in situ* probe from the locus, the $aqz^{GFP}$ GFP expression in heterozygotes, and the anti-Aqz antibody, meaning that transcription and translation are not differentially regulated. This coincidental expression of GFP::Aqz with the anti-Aqz antibody staining in $aqz^{GFP}$ heterozygotes can be used as a marker for Aqz expression (Fig. 4). $aqz^{GFP}$ homozygotes have a clumped, abnormal accumulation of GFP::Aqz, in many cases appearing nuclear. This altered expression serves as a marker for homozygous mutant embryos. During

early embryogenesis in the syncytial blastoderm (st. 5 in Fig. 4), expression is seen in the peripheral cytoplasm and centrally located nuclei of future vitellophages. In later stages, during germband extension/retraction and dorsal closure stages, the ectoderm is heavily labeled, particularly the neuroectoderm. There is also mesodermal Aqz labeling (st. 9–10 and 14 in Fig. 4). During germband retraction, the peripheral nervous system (PNS) and ectoderm are labeled, and some amnioserosa cells nuclei (st. 14 in Fig. 4). As the NS condenses and embryogenesis finishes, the NS expresses *aqz* again, whereas the epidermal tissue expression diminishes (st. 17 in Fig. 4). This expression pattern is consistent with *aqz* requirements in ectoderm and ectodermally derived embryonic tissues.

### *aqz* ectodermally-derived mutant phenotypes

Two previous genetic screens identified *aqz* during embryogenesis. Mutant *aqz* germline clones generated dead embryos with anterior and dorsal holes (*Perrimon et al., 1996*). RNAi injection in developing wild type embryos lead to underdeveloped commissures in the ventral nerve cord, resulting in widening of the distance between longitudinal connectives of the ventral nerve cord commissures (VNC), and severe hypoplasia of the PNS (*Ivanov et al., 2004*). In both cases, end-of-embryogenesis phenotypes were evaluated and not quantitated. Despite possible caveats (e.g., multiple mutant hits on the chromosome arm homozygosed for germline clones generation, and putative specificity problems of RNAi, etc.) mutant phenotypes strongly implied multiple *aqz* requirements in embryonic ectodermally derived tissues.

We used 22C10 and BP102 antibodies to examine nervous system commissures and the PNS, and anti-Elav plus anti-coracle antibodies to examine post mitotic neurons and epidermis, respectively. In late stage wild type embryos, there is co-expression of Coracle and Aqz proteins in the epidermis, especially in the anterior compartment of every segment (Fig. 5A). In contrast, there is no overlap with Elav staining (Fig. 5B), but there is a clear overlap in some somas of the PNS throughout the body (Fig. 5C). Finally, there is no overlap with Repo protein in glial cells (Fig. 5D). We examined embryonic defects in $aqz^{GFP}$, $aqz^1$, and other *aqz* mutant alleles. We wanted to know whether results reported for embryonic RNAi injections and germline clones would be similar. We corroborated and extended the *aqz* mutant cuticles and the disarray of the NS findings reported previously (*Ivanov et al., 2004*; *Perrimon et al., 1996*). We first quantified the number of $aqz^{GFP}$ mutant homozygotes that die during embryogenesis. We separated and cultured homozygous mutant $aqz^{GFP}$ embryos alongside wild type Oregon R control embryos, and found that 65.5% ($n = 417$) die as embryos, before hatching, whereas only 20.8% Oregon R embryos fail to hatch ($n = 173$). All $aqz^{GFP}$ mutant larvae fail to reach adulthood. A percentage of $aqz^{GFP}$ mutant embryos exhibit lack of or abnormal commissures and a strong reduction of the PNS, and sport dorsal, lateral and/or ventral holes in the cuticle (Figs. 3A–3F). $aqz^1$ mutant embryos show the same phenotypes (Figs. S3A–S3F). All other alleles examined showed similar mutant phenotypes (Fig. S2). Mutant embryos also exhibit neurogenic-like phenotypes and cell polarity defects, consistent with Aqz being required in the developing ectoderm (see below).

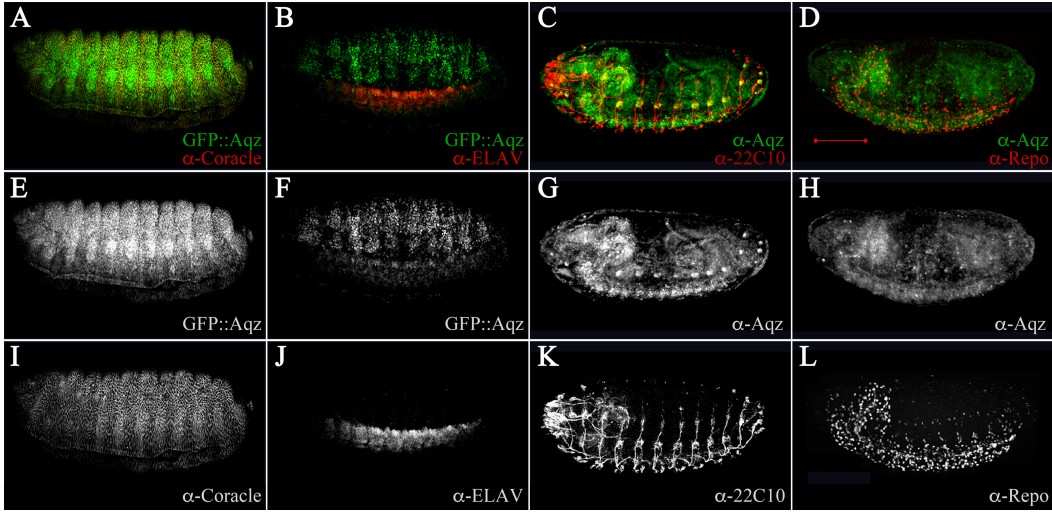

**Figure 5** **Aqz expressed in ectodermally derived tissue.** (A) shows an anti-GFP staining for GFP::Aqz expression together with an epidermal marker, anti-Cora, showing some coincidental expression (yellow). For all immune-stainings in the figure, the panels marked underneath the top A–D labeled E–F show the GFP::Aqz channel separate, and the G–H show the anti-Aqz. The bottom row of panels marked with I, J, K and L depict the anti-coracle (I), anti-Elav (J), anti-22C10 (K), and anti-repo (L) channels, respectively. (B) shows anti-GFP for GFP::Aqz and anti-Elav, to mark neuronal cells in the developing central NS. (C) Using monoclonal 22C10, there is coincidental expression with anti-Aqz antibody, showing that some neurons in the PNS are positive for both Aqz and 22C10 (yellow). (D) In contrast no glial cells (marked by anti-Repo antibody) are coincidental with anti-Aqz antibody staining, showing that glial cells do not express Aqz. Scale bar is 100 µm.

Our expression results, the germline clones, and the RNAseq data from modEncode all suggest an *aqz* maternal contribution partly fulfilling early Aqz requirements. These data firmly establish an *aqz* requirement for ectoderm development and differentiation. Since these are the first *aqz* mutant phenotypes, we focused on these embryonic ectodermal phenotypes, as they may help explain later ones.

### *aqz* NS phenotypes

We wanted to study when are NS defects first seen in *aqz* mutants. To study this, we stained mutant embryos at different stages with different markers of the differentiating NS (Fig. 6). *aqz*[1] mutant embryos does not seem to have a different number of Achaete-positive staining cells (Figs. 6A compared to 6B). Later, during germband extension/retraction, *aqz*[1] mutant embryos have a non-significant, but higher number of Deadpan-staining cells, that labels delaminating neuroblasts (Figs. 6C compared to 6D). *aqz*[1] heterozygotes have $121 \pm 15.4$ Deadpan-positive cells per embryo ($n = 12$), while *aqz*[1] mutant homozygotes have $157 \pm 22$ Deadpan–positive cells per embryos ($n = 16$). *aqz*[GFP] mutant embryos, also stained with Deadpan antibodies, show irregular staining. Some embryonic areas show increased density of Deadpan positive cells, whereas other areas show a dearth of labeled cells (Figs. 6I compared to 6J). Cells are irregularly spaced, with some areas of the embryo showing ectopic stained cells (Figs. 6I compared to 6K, 6L). In a complementary fashion, in the mutant, clumped Aqz protein accumulation (in this instance the chimeric GFP::Aqz)

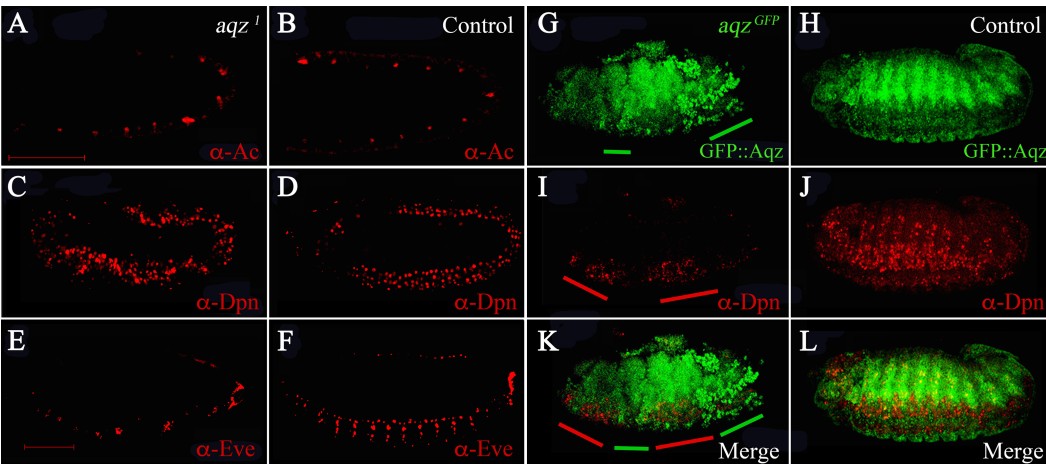

**Figure 6 *aqz* NS defects are detected from the onset of NS formation.** (A, C, E and G) are *aqz* homozygous mutant embryos stained with markers for the NS. (B, D, F and H) are corresponding heterozygous embryos (controls). All views are lateral with anterior to the left, dorsal up. A and B show anti-Achaete staining marking proneural cell during germband extension/retration. C and D show deadpan marked neuroblasts during germband extension/retraction and condensation of the NS. Notice irregular appearance of neuroblasts in C compared with D (control). E and F show expression of Evenskipped (Eve), marking ganglion mother cells, showing a disruption in the mutant (compare E to F) at this later stage of development (stage 15). G and H mark expression of GFP::Aqz. Green lines underneath G mark parts of the embryo with abnormal GFP::Aqz expression, compared to H, where the GFP::Aqz signal is distributed homogeneously. I marks neuroblasts. They also show a clumped and irregular distribution (compare to J which shows a regular array of neuroblasts). Red lines mark areas with abundant neuroblasts in I. K and L show the merged images. Scale bar is 100 μm.

occurs where there is little or no Deadpan staining (Figs. 6G compared to 6I). A similar phenotype is seen in *aqz*[1]: Deadpan positive cells are irregularly spaced (compare Figs. 6C with 6D). Aqz expression increases early before neuroblast delamination, and decreases after the neuroblasts have delaminated, and start dividing (Fig. 4). In *aqz*[1], ganglion mother cells are much reduced and irregularly spaced, as labeled by Anti-Eve antibodies (Figs. 6E compared to 6F). This may mean that some mutant neuroblasts fail to properly differentiate to ganglion mother cells, or differentiate later, a rationale that may explain the tendency of an increased number of neuroblasts.

By stage 16, Aqz is expressed in the epidermis (Fig. 5A). Using epidermal markers, some parts of the ectoderm show lack of staining (Figs. 7A–7I). In these areas, there are more Deadpan-positive cells, suggesting ectopic neuroblasts have formed at the expense of epidermal precursors (Figs. 7A–7I compared to 7O–7S). This could explain the lack of cuticle in some parts, and as a consequence, the cuticular holes found in dead embryos, strongly suggesting a neurogenic phenotype.

## Aqz, Notch pathway and epithelial polarity

The Notch pathway regulates neuroblast and epidermoblast numbers and specification via lateral inhibition during ectoderm differentiation (*Campos-Ortega & Knust, 1990*; *Hartenstein et al., 1992*). *Aqz* mutants show perturbed neurogenesis and epidermogenesis, with phenotypes akin to Notch mutants, i.e., neurogenic phenotypes.

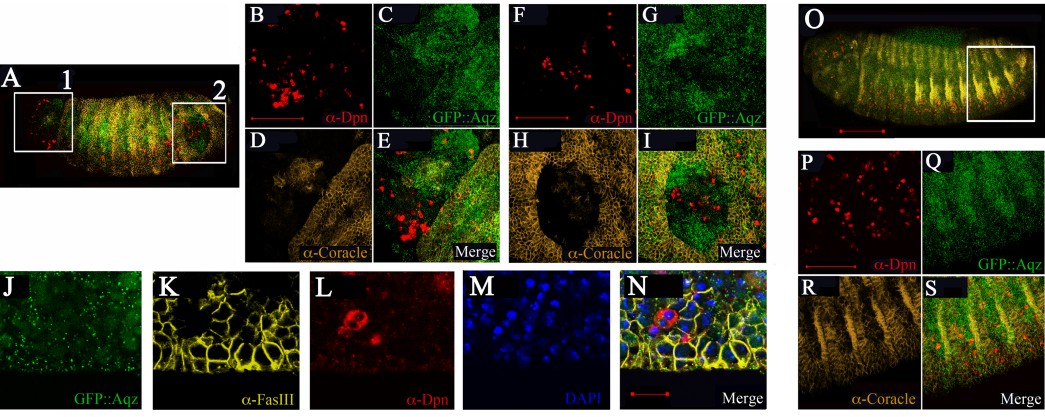

**Figure 7** **Mutations in Aqz lead to irregular distribution of epithelial tissue and NS.** A and O show lateral views of embryos (anterior to the left and dorsal up) stained for neuroblasts (Deadpan antibodies, red), epithelia (Coracle antibodies, yellow), also showing expression of Aqz (green). Scale bar is 100 μm. In A, two distinct regions showing irregular distribution of labels: A1 and A2, are enclosed in white rectangles and enlarged to the right. For comparison, a similar control area is shown in O, the heterozygous control embryo. B, F and P show enlarged anti-Deadpan staining (red); C, G and Q show enlarged GFP::Aqz expression, D, H and R show enlarged anti-Coracle expression, whilst E, I and S show merged images, respectively. Scale bar is 50 μm. J–N show another example of lack of epithelial cells using a different epidermal marker (anti-FasIII, K) and an irregular distribution of neuroblasts (anti-Deadpan, L) in $aqz^{GFP}$ mutant embryos (GFP::Aqz, J). Nuclei are labeled by DAPI (M). N show the respective merged image.

How can Aqz affect Notch signaling? One possibility is that Aqz alters epithelial cell polarity, which is known to affect the apical localization of Dl and consequently, its interaction with the Notch receptor (*Krahn & Wodarz, 2009*). In order to study epithelial polarity, we used antibodies against different known epithelial polarity markers in $aqz^{GFP}$ embryos. We used anti-Crumbs antibodies (anti-Crb) as an apical domain marker (Figs. 8A–8H), and anti-Discs large antibodies (anti-Dlg) as a basolateral marker (Figs. 8I–8P). Results show that the apical domain is compromised, showing reduced Crumbs staining, whereas the basolateral domain is expanded (Discs large staining is augmented).

We then asked whether adherens junctions and Delta expression are altered. We used DE-cadherin antibodies as an adherens junction marker. Adherens junctions are also compromised, since DE-cadherin shows significantly reduced and patchy staining in $aqz^{GFP}$ homozygotes, and so does *Dl* expression, which is also reduced and patchy (Figs. 8Q–8X).

Altogether, the altered epithelial polarity and consequent reduced Dl expression, may explain the neurogenic phenotypes associated with *aqz* mutants, and the ectodermally derived *aqz* phenotypes. This implies cell polarization defects starting at neurulation being the ultimate cause of *aqz* mutant defects and lethality, and provide an explanation for *aqz* involvement in ectodermally derived tissues. It would be interesting to investigate whether other epithelia at other stages also bear similar defects.

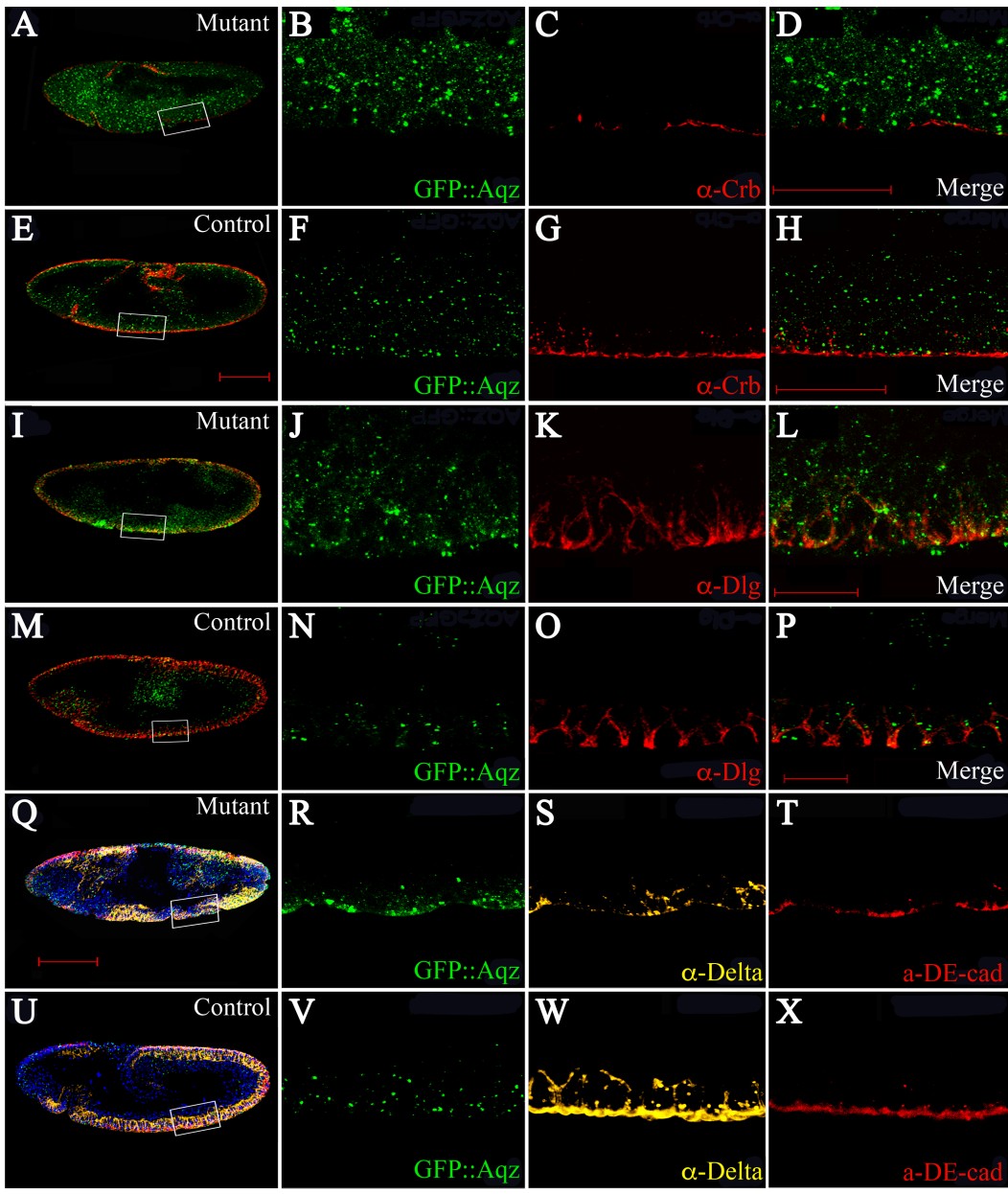

**Figure 8** **The *aqz* mutant neuroepthelium has polarity defects.** A, E, I, M, Q, and U are lateral views of embryos undergoing dorsal closure, with anterior to the left and dorsal up. Bars are 100 μm. A–D, I–L, and Q–T are *aqz*$^{GFP}$ mutant embryos, whereas E–H, M–P, and U–X are *aqz*$^{GFP}$ heterozygtes as controls. B, F, J, N, R, and V show GFP::Aqz expression, C and G anti-Crumbs (Crb) (red), K and O anti-Discs large (Dlg) (red), and S and W anti-Delta (Dl) (yellow). D, H, L, and P show respective merged images. Scale bar is 50 μm. T and X show anti-DE-cadherin (DE-Cad) (red). Note that homozygous mutant GFP::Aqz expression appears more clumped compared to the heterozygous controls, as stated before, and aids in identification of mutant embryos.

## DISCUSSION

*aaquetzalli* is a complex locus encompassing at least three different mRNAs and three different protein products. They code for proteins with a conserved domain, the PNRC, of which *D. melanogaster* harbors two genes. Vertebrate genes with such a domain function in nonsense-mediated RNA decay proteins, and interact with nuclear receptors of varied function. What does Aqz do? Having a second gene with the same conserved domain might mean they are redundant, or partially redundant, and that some functions might be masqued by the activity of the other, even if they diverged somewhat and acquired new functions. Coding for multiple proteins also means that unless we mutate all forms (like in *aqz*$^{GFP}$), we might not get a 'pure' loss-of-function phenotype, and instead have complicated alleles that may be at the same time loss and gain of function. It also leads to complex complementation patterns and phenotypes. Regardless, we observe that most of the phenotypes are consistent, and, in embryos, affect the ectoderm and ectodermally derived tissues.

Why is the phenotype partial in embryos? We know penetrance is not 100% (not all mutant embryos die). We reason that besides the second PNRC-containing gene alluded to above, there is maternal contribution, that may obscure early mutant phenotypes. In *aqz*$^{GFP}$ adults are never seen, with death occurring during a prolonged phenocritical period from embryogenesis to pupariation. From this, we sought to study the earliest defects, in order to gain insight into the causes originating the mutant phenotypes. We found that these phenotypes start very early, around cellularization and cell polarization.

We also show here that Aqz is expressed throughout the life cycle, and is pleiotropic. Yet we find it works in cell polarity early in development, setting the stage for organ differentiation in ectodermally derived tissues, like the epidermis and the NS. We also show that the common thread for these defects is cell polarization, in particular, loss or reduction of the apical domain. This, in turn, leads to compromised Dl and adherens junctions (AJ) localization and abundance (evidenced by Dl and E-Cad staining, respectively), and a neurogenic-like phenotype.

Other genes, besides Notch genes, have neurogenic phenotypes. Some have cell polarity defects, like the cell polarity regulatory proteins (CPRPs), that promote distribution of cellular components (i.e., E-cadherin, a component of AJ, and Dl) in asymmetric patterns resulting in polarized cells (*Sasaki et al., 2007*). Abnormal conditions in these genes, by way of cell polarity defects, may lead to the defects seen in *aqz* mutants. The way in which they cause these defects is currently unknown, but would be interesting to see whether they interact with Aqz in flies, or an Aqz-like vertebrate protein, as Aqz behaves much like them.

## CONCLUSIONS

In this paper we have studied the embryonic epithelial defects of *aqz* mutants, specially in the embryonic ectoderm, and ectodermally derived tissues. We have found that there is a defective neural/epithelial development and differentiation, and that these defects could be ascribed to faulty epithelial polarization. It remains to be seen whether *aqz*, being

transcribed at all stages of the life cycle, is similarly required for other epithelial polarization processes, like in the compound eye or in the wing epithelia. It also remains to be seen whether the *aqz* maternal contribution is requied earlier in development, and whether the other PNRC-domain-containing protein in the fly genome indeed is partially redundant with Aqz function.

## ACKNOWLEDGEMENTS

We would like to acknowledge the assistance of Peña-Rangel.

### Funding

This work was supported by Consejo Nacional de Ciencia y Tecnología of Mexico, CONACYT (No. 177962), the Programa de Apoyo a Proyectos de Investigación e Innovación Tecnológica de la Universidad Nacional Autónoma de México, PAPIIT (No. IN203110), and laboratory budget to Juan R. Riesgo-Escovar. Miguel Ángel Mendoza Ortíz was a doctoral student from Programa de Doctorado en Ciencias Biológicas, Universidad Nacional Autónoma de México (UNAM) with fellowship #164428 from Consejo Nacional de Ciencia y Tecnología (CONACYT). The funders had no role in study design, data collection and analysis, decision to publish, or preparation of the manuscript.

### Grant Disclosures

The following grant information was disclosed by the authors:
Consejo Nacional de Ciencia y Tecnología of Mexico, CONACYT: 177962.
Programa de Apoyo a Proyectos de Investigación e Innovación Tecnológica de la Universidad Nacional Autónoma de México, PAPIIT: IN203110.

### Competing Interests

Juan R. Riesgo-Escovar is an Academic Editor for PeerJ.

### Author Contributions

- Miguel A. Mendoza-Ortíz conceived and designed the experiments, performed the experiments, analyzed the data, contributed reagents/materials/analysis tools, prepared figures and/or tables, authored or reviewed drafts of the paper.
- Juan M. Murillo-Maldonado and Juan R. Riesgo-Escovar conceived and designed the experiments, performed the experiments, analyzed the data, contributed reagents/materials/analysis tools, prepared figures and/or tables, authored or reviewed drafts of the paper, approved the final draft.

### DNA Deposition

The following information was supplied regarding the deposition of DNA sequences:
The sequences are available in Fig. S1.
## Data Availability

The raw data are provided in the Supplemental Files.

## Supplemental Information

Supplemental information for this article can be found online at http://dx.doi.org/10.7717/peerj.5042#supplemental-information.

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
