# Peer review of "aaquetzalli is required for epithelial cell polarity and neural tissue formation in Drosophila"

_PeerJ, doi:10.7717/peerj.5042_

## Round 0.1 · original submission · Minor Revisions

Please address the issues raised by both reviewers on figure labeling, typos and sentences and if possible to use staining with antibodies for clarity.

·

Basic reporting

The article is well written and only contains few typos and sentences that should be corrected/improved.
line 31-32: “aqz encodes pioneer proteins with a conserved domain.” is vague and it would be better that the abstract clearly states the findings of the article: “aqz encodes proteins that harbor a domain with significant homology to a proline-rich conserved domain of nuclear receptor co-activators.”
line 91: “RNAi injected in developing embryos sport defects…” should be replaced by “RNAi injected in developing embryos show defects…”
line 146: “deoxycytidine 5’-trisphoshpate” should be corrected to “deoxycytidine 5’-triphosphate”
Line 466:”…differentiate to ganglion mother cells, r differentiate later, a rationale…” should be corrected to ”…differentiate to ganglion mother cells, or differentiate later, a rationale…”

Figure 1 legend states that the LD47990 was used as a probe for aqz-RA while LD02060 is used for RB, RC and RD. The location of LD02060 is indicated on the figure (showing it cannot detect RA transcripts) but the location LD47990 is missing to confirm that the probe only detects the RA transcript.

The material and method section for Northern-blot is in disagreement with figure 1 by stating that probes were from LD47990 and the aqz 5’ region instead of LD02060 (line 145).

The description of the aqzRescue construct is not provided in the material and method (cloning procedure, injection, transgenic strains) or indicated on Figure 1. Figure indicates rescue on the map that corresponds to a BAC construct that contains many transcription units that would not allow the authors to conclude that the lethality and neurogenic phenotypes are caused by the CG9821 transcription unit.

Figure 3G and 3H genotypes are incorrectly reported: the GFP allele is not homozygous in the rescue genotypes and appears located on 2 different chromosomes (using ; instead of ,). The rescue construct is on the 2nd (53B2) and the aqz allele on the third (85B2). The genotypes should be aqzRescue/+; aqzGFP (1 copy rescue) or aqzRescue;aqzGFP(2 copies rescue). Homozygous genotypes should also be written aqzGFP and aqzRescue (no need for /).

Figure 3H (and supplemental Figure 3H) is very confusing and hard to understand. It appears as almost 100% of the wild-type control die as adult. Bar graphs showing the lethality at each stage (3 bars) as % of previous stage would greatly help.

Experimental design

The authors needs to provide more details on how the Northern-blot and in situ probes were made (clone used and fragment used for multi prime labelling). The amount of RNA used for the Northern-blots is also missing.

Validity of the findings

It is pretty rare that a gene is poorly documented on FlyBase and that the nature and number or transcripts/proteins solely result from computer predictions. The authors finally provide experimental data that clearly show the inaccuracy of the computer predictions regarding the size and number of transcripts. The use of the Northern-blot technique provides a much more accurate examination of the locus that does not involve any PCR or reverse transcription steps that can easily distort the results. The authors should however indicates (and ideally provides in supplementary data) that the probe has been confirmed to only detect transcripts from the aqz locus (the probe used is not strand-specific) by using single-stranded probes or showing absent/reduced northern signal on RNA isolated from mutant alleles.

The rescue experiments with the aqzRESCUE construct are the most conclusive data that the neurogenic phenotype of the aqz1 and aqzGFP alleles are caused by an alteration of the CG9821 transcription unit. The complementation data (Table 1) show that the lethality phenotype is complemented between aqz1 and aqzGFP (confirmed by fig 3H and sup Fig 3H: most lethality occurs in larvae with aqzGFP while most lethality occurs in pupae with aqz1). This observation implies that aqz1 and aqzGFP cause lethality by altering different genes. It is confirmed by deficiency complementation: aqzGFP lethality is not complemented by the deficiencies (therefore the alteration is located between 85B1 and B6 as stated in Table 1 legend) while aqz1 lethality is complemented therefore indicating that the defect causing the lethality is not located between 85A5 and 85C2. The authors did not test the rescue of the legality of the 2 alleles similarly (no wild-type control and no rescue with only one copy of aqzRescue) but it seems that the pupal lethality is comparable between homozygous aqz1 and the rescue genotype (~50% larvae die and 100% of the 50% larvae that pupated die, comparable to ~90% pupal lethality in homozygous, larval lethality is likely due to the use of 2 copies of aqzRescue as observed in Figure 3). It is therefore most likely that the aqz1 strain has 2 different alterations: one outside of the aqz region that cause pupal lethality and one within the region that is not strong enough to cause larval lethality (or very weak considering that the larval lethality appears a bit higher than the wild-type control used in aqzGFP rescue in Figure 3) but do cause neurogenic phenotypes. It is critical for the validity of the findings that the authors describe the aqzRESCUE construct used to show that it only contains the aqz transcription unit. The rescue construct shown in Figure 1 (BAC) contains many transcription units and would not allow concluding that the neurogenic phenotypes are caused by the CG9821 transcription unit.

It is unfortunate that the authors were unable to obtain antibodies. The authors do not have any experimental data to demonstrate that CG9821 transcription unit encodes more than one kind of protein. The abstract (line 31-32) should say that the locus encodes a putative protein (no plural).

Reviewer 2 ·

Basic reporting

Results are clearly presented and well explained except at some places like
1. Fig.5, it would be more clear if you also provide the separate channel images along with the merged one.
2.There are some typo and spacing errors, one again a thorough reading will solve that.
3. Fig.8 has lots of discrepancy in labeling.

Experimental design

1.Expression pattern of Aqz would be more clear with co-immunostaining for specific neuroectodermal, peripheral nervous system or mesoderm in Figure 4. Can show with any one of them which shows the best expression.
2. What about the Aqz expression in Aqz mutant, can show with alpha- Aqz antibody staining.
3. Fig.8, Mutant embryos are showing more intense expression of GFP-Aqz as compared to control embryos, is it really the case or just because of merging of multiple stacks during processing in case of mutant or if there is more intense expression then why it is so.
Or It would be more better to use Aqz antibody instead of GFP.

Validity of the findings

no comment

Additional comments

This article further enhances our understanding about the role of Aqz during the development specially in the embryonic ectoderm, and ectodermally derived tissues. Moreover, this study indicates toward its even more major involvement in various important developmental processes by altering the expression of polarity proteins and that would be even more interesting to study.

---

## Round 0.2 · accepted · Accept

Authors have address all comments raised by both reviewers. Revised manuscript is ready for publication.

#